# Cellulitis Is Associated with Severe Breast Cancer-Related Lymphedema: An Observational Study of Tissue Composition

**DOI:** 10.3390/cancers13143584

**Published:** 2021-07-17

**Authors:** Mads Gustaf Jørgensen, Anne Pernille Hermann, Anette Riis Madsen, Steffanie Christensen, Kim Gordon Ingwersen, Jørn Bo Thomsen, Jens Ahm Sørensen

**Affiliations:** 1Department of Plastic Surgery, Odense University Hospital, 5000 Odense, Denmark; Joern.Bo.Thomsen@rsyd.dk (J.B.T.); jens.sorensen@rsyd.dk (J.A.S.); 2Clinical Institute, University of Southern Denmark, 5000 Odense, Denmark; 3OPEN, Open Patient data Explorative Network, Odense University Hospital, 5000 Odense, Denmark; 4Department of Endocrinology, Odense University Hospital, 5000 Odense, Denmark; Pernille.Hermann@rsyd.dk (A.P.H.); Anette.Riis.Madsen@rsyd.dk (A.R.M.); steffanie.christensen@rsyd.dk (S.C.); 5Research Unit in Physiotherapy and Occupational Therapy, University Hospital of Sourthen Denmark—Vejle Hospital, 7100 Vejle, Denmark; Kim.Ingwersen@rsyd.dk; 6Department of Regional Health Research, University of Southern Denmark, 5000 Odense, Denmark

**Keywords:** lymphedema, bio composition, DEXA, lymphangiography, bioimpedance, fat, fluid, excess, proportion

## Abstract

**Simple Summary:**

Cellulitis is a common complication in Breast Cancer-Related Lymphedema (BCRL); however, it is not known whether cellulitis is associated with the severity and biocompositon of BCRL. This study showed that cellulitis was associated with more excess volume, fat, and lean arm mass. Treatments should aim to prevent cellulitis in BCRL to possibly avoid the condition exacerbating.

**Abstract:**

Cellulitis is a common complication in Breast Cancer-Related Lymphedema (BCRL). The excess amount of fat and lean mass in BCRL is a vital factor in patient stratification, prognosis, and treatments. However, it is not known whether cellulitis is associated with the excess fat and lean mass in BCRL. Therefore, this prospective observational study was designed to fundamentally understand the heterogonous biocomposition of BCRL. For this study, we consecutively enrolled 206 patients with unilateral BCRL between January 2019 and February 2020. All patients underwent Dual-Energy X-Ray Absorptiometry scans, bioimpedance spectroscopy, indocyanine green lymphangiography comprehensive history of potential risk factors, and a clinical exam. Multivariate linear and beta regression models were used to determine the strength of association and margins effect. Sixty-nine patients (33%) had at least one previous episode of cellulitis. Notably, a previous episode of cellulitis was associated with 20 percentage points more excess fat and 10 percentage points more excess lean mass compared to patients without cellulitis (*p* < 0.05). Moreover, each 1 increase in the patients BMI was associated with a 0.03 unit increase in the fat mass proportion of the lymphedema arm. Cellulitis was associated with more excess fat and lean arm mass in BCRL. In addition, patients BMI affect the proportion of fat mass in the arm.

## 1. Introduction

Arm lymphedema is one of the most common and feared side effects of breast cancer treatment with lymph node involvement and affects up to one in three breast cancer survivors [1,2]. The cause of breast cancer-related lymphedema (BCRL) is the obstruction of normal lymph circulation [3]. BCRL is associated with increased fluid and fat accumulation, and patients are at an increased risk of developing episodes of cellulitis [4]. Conservative treatments such as compression and microsurgical treatments aim to drain the fluid component of BCRL. However, both of these treatments have shown inconsistent results [5,6], which can be due to interindividual differences in the excess amount of fat and fluid in the arm [7,8]. Cellulitis is a soft tissue infection and a common complication in BCRL, which has been associated with more excess swelling and a higher dependency on conservative treatments [9,10]. Cellulitis in BCRL has been thought to damage lymphatic vasculature and further exacerbate BCRL [11,12], however clinical data have been lacking [13]. The characteristics of BCRL patients who have previously had an episode of cellulitis are not well understood, despite being a large patient subgroup with a worse prognosis [9,10]. In light of recent progress in BCRL treatments, the biocomposition of BCRL is set to become a vital factor in patient stratification, prognosis, and treatment-decision making between conservative, microsurgical or debulking treatments [14,15,16]. Therefore, it is critical to assess if BCRL patients with a previous episode of cellulitis have a different biocomposition profile to BCRL patients without cellulitis.

This study was designed to understand the fundamentals and heterogeneity of the clinical presentation of BCRL through investigation of a large patient group undergoing extensive biocomposition analysis. The primary aim of this study was to investigate the association between a previous cellulitis episode and the amount of fat mass in the lymphedema through Dual-Energy X-Ray Absorptiometry (DEXA) scans and bioimpedance spectroscopy analysis. Second, we aimed to investigate factors associated with a higher proportion of fat mass in the lymphedema arm.

We hypothesized that patients with a previous cellulitis episode would have more excess fat mass in the lymphedema arm compared to patients without a previous cellulitis episode.

## 2. Materials and Methods

### 2.1. Study Design and Setting

This was a cross-sectional study reported in accordance with the Strengthening the Reporting of Observational Studies in Epidemiology (STROBE) Statement for cross-sectional studies [17]. Patients were consecutively enrolled for this study at the department of Plastic and Reconstructive Surgery, Odense University Hospital between January 2019 and February 2020.

### 2.2. Participants

The patients included in this study were all referred from outside hospitals, general practice, or self-referred to our department for experimental lymphedema treatment. We defined BCRL clinically as a history of BCRL diagnosis by a certified lymphedema physiotherapist, in agreement with the International Society of Lymphology guidelines [16]. In addition, all patients had to adhere to conservative compression treatments for at least one year prior to study enrollment, to rule out intra-personal fluctuations in BCRL presentation. All BCRL patients were screened for study eligibility per email and telephone and were invited explicitly for study participation based on the following criteria:Previous treatment for loco-regional breast cancer with axillary lymph node dissectionCancer-free for more than one yearBody mass index ≤ 35American Society of Anesthesiology score of 1 or 2 [18]Able to give informed consentNo history of other malignancy apart from breast cancer and non-melanoma skin cancerNo insulin-dependent diabetesNo known hepatitis, HIV, or syphilis infectionNo primary lymphedema or non-breast cancer-related lymphedema

Eligible patients signed the informed consent form and had to undergo a 2 h study assessment at the department of Plastic and Reconstructive Surgery, Odense University Hospital. The study assessment consisted of DEXA scans, bioimpedance spectroscopy, indocyanine green lymphangiography (ICG-L) of lymphatic flow, and clinical assessment conducted in that order.

### 2.3. Outcome Measurements

#### Dual-Energy X-ray Absorptiometry

Dual-Energy X-ray Absorptiometry is a scan modality that measures bone mass, fat mass, and lean mass (e.g., fluid and muscle) with high precision and accuracy [19,20]. All DEXA scans were performed at daytime using the Discovery/Horizon A densitometers (Hologic, Waltham, MA, USA, serial number 82245/301872M) and analyzed using the Hologic APEX software version 13.3:3/13.6.0.5:3. All participants were positioned supine on the scan table with their head positioned at the table’s top border. To capture both the lymphedema and healthy arms, two whole-body DEXA scans were performed on each patient (one for each arm). Before each scan, patients were moved laterally from the table’s midline, with the elbow and shoulder extended at approximately a 90-degree angle to avoid an overlap with the torso and thighs. The patient’s hands were positioned palmar side down with extended and separated fingers. Before the appointment, patients were instructed to dress lightly before each scan. The scans were subsequently analyzed using arm sub-regions. The region of interests was drawn manually for each arm extending from the fingertips and 25 software-standardized units proximally. For both arm regions, the amount of fat mass, lean mass, and bone mass was automatically calculated in grams by the software, and the volume was subsequently derived using standardized densities [21,22].

### 2.4. Bioimpedance Spectroscopy

Bioimpedance spectroscopy is a noninvasive technique that measures extracellular fluid in both the lymphedema and healthy arm and whole-body composition variables. Extracellular fluid and whole-body composition variables are measured by facilitating a small alternating electric current through the extremities and the impedance of the electric current is then retrieved at multiple frequencies (range: 3–1000 kHz). Patients in this study underwent bioimpedance spectroscopy using a valid and reliable stand-on device (SOZO™, Impedimed, Carlsbad, CA, USA) [23]. Before each bioimpedance reading, patients were asked void their bladder and to remove all jewelry. Before the appointment, patients were further asked to refrain from excessive physical activity, caffeine, and alcohol consumption 24 h before the assessment per the manufacturer’s instructions. The extracellular fluid and whole-body composition parameters were then derived from the impedance values using the SOZOthrive (SOZO™, Impedimed, Carlsbad, CA, USA) and SOZOpro (SOZO™, Impedimed, Carlsbad, CA, USA) software algorithms after inspection of the cole plots. All measurement data are stored in the secure SOZOcloud (SOZO™, Impedimed, Carlsbad, CA, USA).

The impedance ratio of the lymphedema and healthy arms is expressed as an L-DEX score using the SOZOthrive software. Normal L-DEX scores for patients without BCRL range between −10 and 10. Patients with BCRL have L-DEX values above 10, signifying more fluid edema in the arm.

The whole-body composition parameters derived from the SOZOpro software included total body water (liters), extracellular fluid (liters), intracellular fluid (liters), fat-free mass (kg), extracellular mass (kg), active cellular tissue mass (kg), fat mass (kg), skeletal muscle mass (kg), the mass of protein and minerals (kg), basal metabolic rate (calories per day), Hy-Dex hydration score, and phase angle (°). Active tissue mass is an estimate of the amount of metabolic tissue with higher values associated with greater health and physical fitness. Hy-Dex hydration score gives an estimate of the individuals fluid status with higher values signifying more bodily fluid. Phase angle is the arctangent of resistance of the persons cell membranes at 50 kHz frequency and is a surrogate for cell membrane function.

### 2.5. Indocyanine Green Lymphangiography

Indocyanine green lymphangiography was used to evaluate lymphatic flow and the severity of lymphatic injury dermal backflow. Indocyanine green lymphagiography was performed by injecting 0.1 mL indocyanine green (2.5 mg/mL Verdye, Diagnostic Green, Ascheim, Germany) subcutaneously and intradermally at the first and third finger webspace and at the wrist near the ulnar border of the palmaris longus tenon of the affected arm as previously described [3]. The entire arm was scanned after 1 h of indocyanine green injections using the HyperEye medical system (MNIRC-501, HEMS; Mizuho Co., Tokyo, Japan) and staged using the MD Anderson indocyanine green lymphangiography staging system [24,25].

The MD Anderson ICG-L staging systems comprise of 6 stages, and the degree of lymphatic injury and lymphatic dermal backflow is graded on a 0–5 scale:Stage 0: Normal linear lymphatics and no dermal backflowStage 1: Many patent lymphatics and minimal lymphatic dermal backflowStage 2: Moderate number of patent lymphatics and segmental dermal backflowStage 3: Few patent lymphatics and extensive dermal backflowStage 4: Dermal backflow involving the handStage 5: No proximal uptake of ICG from the injection site

Following ICG injection, we monitored all patients for at 1 h for allergic and hypersensitive reactions to the dye.

### 2.6. Clinical Examination

The clinical examination consisted of a detailed patient history, height and body weight measurements, and a clinical lymphedema pitting test. The following demographic information was registered for each patient: Age at assessment (years), marital status (yes/no), employment status (yes/no), time of lymphedema diagnosis (date), previous episode of arm cellulitis (yes/no), lymphedema in dominant arm (yes/no). Furthermore, we specifically asked about the use of first-line conservative lymphedema treatments: compression sleeve (yes/no) and compression gauntlet (yes/no) and second-line conservative treatments: night compression (yes/no), and pneumatic compression devices (yes/no). The patient’s current weight and height were measured in the outpatient clinic, and the body mass index was calculated. The pitting test was performed by pressing the thumb firmly on the lymphedema arm at several points for up to 1 min [26]. Pitting edema was defined as the presence of clinical pitting following the pitting test.

### 2.7. Danish Breast Cancer Registry

The following information regarding previous breast cancer treatment were retrieved from the Danish Breast Cancer Group registry [27]: Time of breast cancer treatment (date), type of breast surgery (mastectomy/lumpectomy), radiation therapy (yes/no), chemotherapy (yes/no), and the number of lymph nodes removed in axillary dissection (number). Regarding the involvement of radiation therapy, we specifically asked patients if the radiation field involved the axilla (yes or no).

The lymphedema latency was defined as the time from cancer treatment until lymphedema diagnosis and was calculated by subtracting these dates. The lymphedema duration was defined as the time from lymphedema diagnosis until study assessment and was calculated by subtracting these dates

### 2.8. Outcome Variables

The primary outcome was the association between a previous cellulitis episode and the amount of excess fat and lean mass in the lymphedema arm, when compared with the healthy arm. Cellulitis was defined as having had at least one previous incident of cellulites in the BCRL arm diagnosed by a physician and treated with oral or intravenous antibiotics.

To account for interindividual variations in healthy arm sizes, the excess fat and lean mass derived from DEXA scans were calculated as percentages using the following equation:(1)$$\frac{Lymphedema\;arm_{mass}-Healthy\;arm_{mass}}{Healthy\;arm_{mass}}\cdot 100$$

The secondary outcome was to investigate factors associated with a higher proportion of fat mass in the lymphedema arm.

The proportion of the lymphedema arm mass and healthy arm mass that was fat, lean, and bone mass was defined as the mass of fat, lean, and bone mass divided by the total arm mass using the following equation:(2)$$\frac{Arm_{mass}}{Arm_{total\;mass}}$$

To account for interindividual variations in patients height and weight, the whole-body composition parameters derived from the SOZOpro software were calculated as percentages using the following definitions: Total body water (as % of body weight), extracellular fluid (as % of total body water), intracellular fluid (as % of total body water), fat-free mass (as % of body weight), extracellular mass (as % of fat-free mass), active tissue mass (as % of fat-free mass), fat mass (as % of body weight), skeletal muscle mass (as % of body weight), the mass of protein and minerals (as % of body weight).

The lymphedema volume was defined as the volume of the affected arm minus the volume of the healthy arm as percentage using the following equation:(3)$$\frac{Lymphedema\;arm_{volume}-Healthy\;arm_{volume}}{Healthy\;arm_{volume}}\cdot 100$$

### 2.9. Statistical Methods

We described the baseline characteristics of patients with means ± standard deviation (SD) for continuous parametric variables, median and interquartile range (IQR) for nonparametric continuous variables, and rounded frequencies (%) for categorical variables. The Skewness/Kurtosis test was used to test for normal distributions of continuous variables. Comparisons between patients with and without cellulitis were done using unpaired *t*-test, Chi-squared, or Mann–Whitney test depending on data type and distribution.

We performed univariate Pearson’s correlation (r) to assess correlation between all variables and interpreted them using the following a priori defined thresholds [28]:

0.00–0.30 = poor correlation

0.31–0.50 = slight correlation

0.51–0.70 = moderate correlation

0.71–0.90 = substantial correlation

0.91–0.99 = near-perfect correlation

1.00 = perfect correlation

Statistically significant variables and clinically relevant variables were included in the multivariate regression models. A multivariate linear regression model was used to analyze the percentage of excess volume, lean mass, and fat mass in the lymphedema arm. A multivariate beta regression model was used to analyze the relationship and margins effect of the proportion of the lymphedema and healthy arm that was fat and lean mass.

STATA 15 (StataCorp. 2017. Stata Statistical Software: Release 15. College Station, TX: StataCorp LP) was used for the statistical analysis and conducted with a two-tailed significance level of 0.05 and reported with 95% CI when applicable.

## 3. Results

We assessed 534 patients and included 206 patients between January 2019 and February 2020 (Figure 1). One-hundred-and-sixty-three patients (79.13%) stated that they had received radiation therapy towards the axilla. The baseline demographics of all included patients are summarized in Table 1. Sixty-nine patients (33.50%) had at least one previous cellulitis episode. Patients with cellulitis had significantly more excess fat mass in their lymphedema arm (median: 37.48%, IQR: 50.53%) compared to patients without a previous cellulitis episode (median: 20.51%, IQR: 35.57%, *p* < 0.05). Furthermore, patients with cellulitis had more excess lean mass in their lymphedema arm (median: 21.30%, IQR: 37.57%) compared to patients without cellulitis (median; 9.13%, IQR: 30.48%, *p* < 0.05). More patients with cellulitis used second-line lymphedema treatments compared to patients without cellulitis (*p* < 0.05). Indocyanine green lymphography was performed in 200/206 patients (97.09%), as the HyperEye was occupated in the operating theater for the remaining six patients (two patients with cellulitis and four patients without cellulitis). The lymphatic injury was significantly worse in patients with cellulitis (Median (IQR) MD Anderson stage, 3 (1), Supplementary VideoAppendix A compared to patients without cellulitis (Median (IQR) MD Anderson stage, 2 (2), Supplementary VideoAppendix A. No significant differences in baseline demographics, breast cancer treatment, or subsequent breast reconstruction were found between patients with and without cellulitis.

### 3.1. Excess Arm Mass

In the univariate analysis, cellulitis was associated with longer lymphedema duration (r = 0.40, *p* < 0.05), excess lymphedema volume (r = 0.28, *p* < 0.05), clinical pitting on clinical exam (r = 0.20, *p* < 0.05), and excess lymphedema fat mass (r = 0.21, *p* < 0.05, Figure 2). Lymphedema in the dominant arm was associated with more excess lymphedema volume (r = 0.41, *p* < 0.05) and more excess lymphedema lean mass (r = 0.38, *p* < 0.05).

The multivariate linear regression analysis revealed that a previous episode of cellulitis was associated with a 12.35 (*p* < 0.001) percentage point increase in excess lymphedema volume, 20.74 (*p* < 0.001) percentage point more excess fat mass and 9.95 (*p* < 0.05) percentage points more excess lean mass (Table 2). Each one unit increase in BMI was associated with 1.44 (*p* < 0.001) percentage points more excess lymphedema volume and 2.18 (*p* < 0.001) percentage points more excess lean mass. In addition, patients with lymphedema in their dominant arm had 13.18 (*p* < 0.001) percentage points excess lymphedema volumes and 17.36 percentage points more excess lean mass.

### 3.2. Fat and Lean Arm Mass Proportions

In the univariate analysis, the lean mass proportion of the lymphedema arm was associated with the lean mass proportion of the healthy arm (r = 0.62, *p* < 0.001, Figure 3A) and negatively associated with the patients BMI (r = −0.34, *p* < 0.001) and body fat mass (r = −0.43, *p* < 0.001). The proportion of fat mass in the lymphedema arm was associated with the proportion of the healthy arms fat mass (r = 0.67, *p* < 0.001), the patients BMI (r = 0.53, *p* < 0.001), and the patient’s body fat mass (r = 0.43, *p* < 0.001, Figure 3B–C). The residuals between the lymphedema and healthy arms fat, lean and bone mass were unbiased and homoscedastic along the x-axis (Figure 3A^1^–C^1^).

The multivariate beta regression analysis showed that for each one unit increase in the patients BMI, there was a 0.02 (*p* < 0.001) decrease in the lean mass proportion of the lymphedema arm, 0.04 (*p* < 0.001) decrease in the lean mass proportion of the healthy arm, 0.03 (*p* < 0.001) increase in the fat mass proportion of the lymphedema arm, and a 0.05 (*p* < 0.001) increase in the fat mass proportion of the healthy arm (Table 3). Similarly, for each increase in patients body fat mass, there was a 0.02 (*p* < 0.001) decrease in the lean mass proportion of the lymphedema arm and healthy arm and a 0.02 (*p* < 0.001) increase in fat mass proportion of the lymphedema and healthy arm (Figure 4).

## 4. Discussions

This study found that a previous episode of cellulitis was associated with more excess fat and lean mass in the arm (Figure 5). In addition, increased BMI was also associated with more excess fat and lean mass in the arm. However, increased BMI and body fat mass was additionally associated with a higher proportion of the BCRL and healthy arm that was fat mass.

This is the first study investigating the association of cellulitis and body composition with BCRL biocomposition, providing framework evidence for the heterogenetic presentation of BCRL. Evidence from this study implies that cellulitis, BMI, body fat mass, and arm dominance are associated with variations in BCRL biocomposition. The study’s strengths include a large number of included patients with representative disease stages and blinded DEXA assessments, lymphography evaluation and comprehensive correlation to bioimpedance and clinical assessment. DEXA measurements, as compared to conventional volume measurements such as tape measurements, water displacement, and perometry, has the advantage of also estimating the volume and mass of the fat and lean content of the arm. The information derived from DEXA assessments, such as the excess lymphedema fat and lean mass content, may be used to guide treatments and patient selection [7]. For example, a patient with a high proportion of excess adipose tissue in the arm may be a poor candidate for treatments aiming at draining excess fluid. In such cases, debulking procedures such as liposuction could be more appropriate in order to remove excess fat [29]. In contrast, a patient with a low proportion of excess fat mass could be a favorable candidate for treatments aiming at draining the excess fluid [30,31]. As expected, we found that BCRL patients had larger excess volumes with more excess fat and lean mass in their lymphedema arm than the healthy arm. This finding substantiates several smaller studies investigating lymphedema bio composition with DEXA scans [7,8,19,32]. This study showed that BCRL is comprised of both a fat and fluid component, therefore combined surgical treatments addressing both the fat and fluid component may be considered in selected patients [33,34,35].

The study’s main weakness is that we could not further divide the lean mass into fluid and muscle. However, this is a dilemma shared between all current lymphedema bio composition measurements, and we believe our conclusions are well supported for the following reasons. Our analysis showed a clear association between lymphedema lean mass and fluid-related variables derived from the bioimpedance spectroscopy measurement. This indicates that a significant proportion of the excess lymphedema lean mass can be attributed to a fluid component and not excess muscle [36]. Magnetic resonance angiography is another compatible technique to assess lymphedema biocomposition, showing the anatomical distribution of the different tissue components and guiding microsurgical lymphedema treatment [15,36]. However, the apparent benefit in this study setting of DEXA over magnetic resonance angiography is that magnetic resonance angiography relies on manual and qualitative assessments. In contrast, DEXA scans provide an exact quantitative measure of each tissue component, which is not currently feasible using magnetic resonance angiography [14,36]. Bioimpedance spectroscopy is increasingly being used for evaluating BCRL due to its quick, simple, and noninvasive design [37]. The L-DEX parameter is the most reported bioimpedance-derived outcome and information have been lacking about the usability of the other fluid parameters such as extracellular fluid and Hy-Dex which have also been reported in BCRL context. In this study, we found that patients with cellulitis had twice as high an L-DEX score as patients without cellulitis. However, there was no difference in the amounts of extracellular fluid or Hy-Dex scores between the groups. This discrepancy can be due to L-DEX measuring the impedance ratio of each upper limb, while extracellular fluid and Hy-Dex are whole-body analysis which also measures the fluid content of the abdomen and thorax, which have a much higher and more variable fluid content compared to the upper extremity [38,39]. This suggests that whole-body bioimpedance spectroscopy is not sensitive enough to assess fluid changes in the upper limbs. Another limitation of the study is that we used observational data and therefore cannot establish causality and only an association between cellulitis and excess lymphedema volume, fat mass and lean mass.

It is an accepted notion that cellulitis in lymphedema can worsen already established lymphedema and increase the risk of future cellulitis episodes [40,41]. This notion has recently been supported by animal simulations showing that bacterial infections corrode lymphatic vessels [42], which inhibits lymphatic function and causing a vicious cycle [43]. We found that patients with a previous cellulitis incident had more extensive lymphatic injury and leakage, as visualized by indocyanine green lymphangiography showing clinical evidence of the damaging correlation of cellulitis and lymphatic vasculature. This hypothesis of cellulitis damaging the lymphatic vasculature is further supported by a large systematic review showing arm cellulitis to also be an independent risk factor for developing lymphedema after breast cancer treatment [13]. Decreased lymphatic transport and lymphatic vessel dilation and sclerosis are pathological hallmarks for lymphedema [44,45]. In fact, cellulitis can deteriorate lymphatic failure and reduce the proximal transport capacity of interstitial fluid [11,12]. Conservative and surgical BCRL treatments aim to enhance lymphatic transport capacity and have shown promising potential for reducing the risk of cellulitis [11,46]. Patients at risk of lymphedema and patients with lymphedema alike should therefore be especially recommended to maintain proper skin integrity to reduce the risk of cellulitis. The worst culprit of BCRL is axillary lymph node dissection, but even minimally invasive sentinel lymph node biopsy carries a risk of BCRL [47,48]. Avoiding axillary lymph node surgery all together by predicting the breast cancer lymph node involvement have the potential to significantly reduce the risk of developing BCRL [49]. Exercise has been shown to reduce the risk of BCRL and breast cancer treatment on the dominant side is not usually associated with an increased risk of BCRL [50,51]. Therefore, it can seem counterintuitive that BCRL on the dominant side was associated with more severe disease, however the dominant arm may be more likely to sustain minor injuries which can predispose to cellulitis and BCRL progression [52]. Few clinical studies have addressed the problem of cellulitis complicating lymphedema [10,53,54]. The few present studies unanimously show that patients with cellulitis have larger excess volumes compared to lymphedema patients without cellulitis. However, the generalizability of their conclusions is limited by the small sample sizes in these studies (<30 patients), mixed lymphedema etiologies (breast cancer, urogenital cancer, head and neck cancer), diverse affected regions (arm, leg, neck) and lack of measurements to assess internal limb bio composition. This present study offers compelling evidence that cellulitis is associated with excess fat and lean mass deposition. This study also showed that lymphedema duration was associated with cellulitis and thereby lymphedema severity, which lend support to increased risk of lymphedema circulus vicious over time. Previous studies have found that abdominal obesity is associated with more excess lymphedema volume [53,55]. This finding is resonated in our analysis showing increased lymphedema morbidity in patients with high BMI and body fat mass. Obesity has previously been associated with worse outcomes following treatments aimed at draining the fluid component of lymphedema [54,56]. We found that increases in BMI and body fat mass was associated with a larger proportion of the lymphedema arm that was fat mass. Because microsurgical and conservative treatments aim at draining the fluid proportion of lymphedema, this may explain why obese patients have worse treatment outcomes. Therefore, BCRL patients may benefit from weight optimization in general and especially prior to surgical interventions [53]. Surprisingly, excess fat and lean mass in the lymphedema arm was not associated with the duration of lymphedema. This is in stark contrast to the generally accepted dogma that lymphedema is initially comprised of a fluid component, that over time is replaced by fat. Yet, similar descriptive studies carried out by Dayan et al. and Brorson et al. support our findings [7,14]. Remarkably, we found that patients with BCRL in their dominant arm had 13 percentage points more excess lymphedema volume (dominated by excess lean mass). This is considerably more than the expected 3% excess arm volume associated with arm dominance in the healthy reference population [22]. Considering the importance of arm dominance, BMI and body fat mass we suggest that patients’ general health status and healthy arm need to be evaluated when selecting patients for treatments and evaluating treatment efficiency.

## 5. Conclusions

Cellulitis was associated with more excess fat and lean mass in the BCRL arm. In addition, the excess amount of fat and lean mass was associated with increased BMI. The proportional amounts of BCRL that was fat and lean mass was further associated with BMI and bodily fat mass. There was a large variation in excess and proportional fat and lean mass, which may lead to the possibility of individualized BCRL treatment.

## Figures and Tables

**Figure 1 cancers-13-03584-f001:**
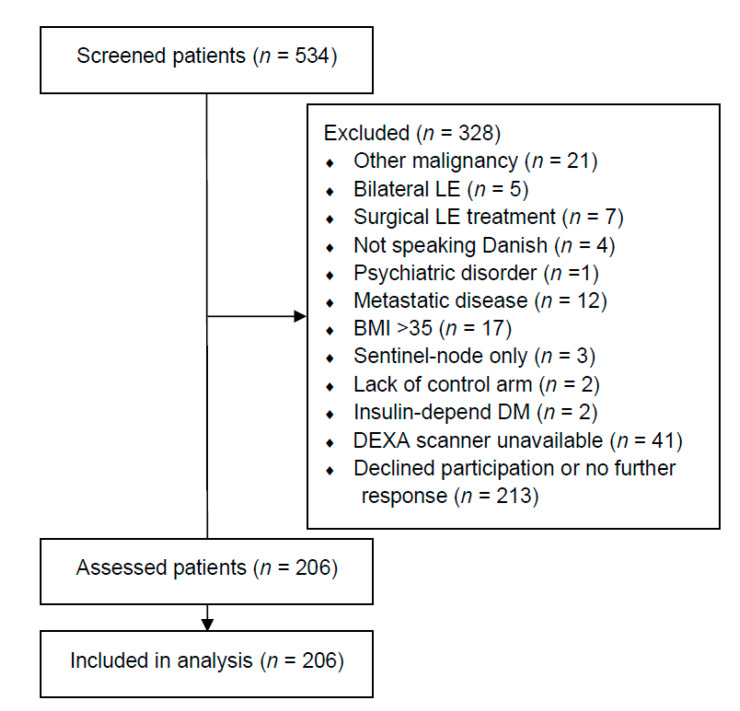
Flow chart of included patients. This table shows the flowchart of the include patients. LE = lymphedema. DM = diabetes mellitus. BMI = body mass index. DEXA = dual-energy X-ray absorptiometry.

**Figure 2 cancers-13-03584-f002:**
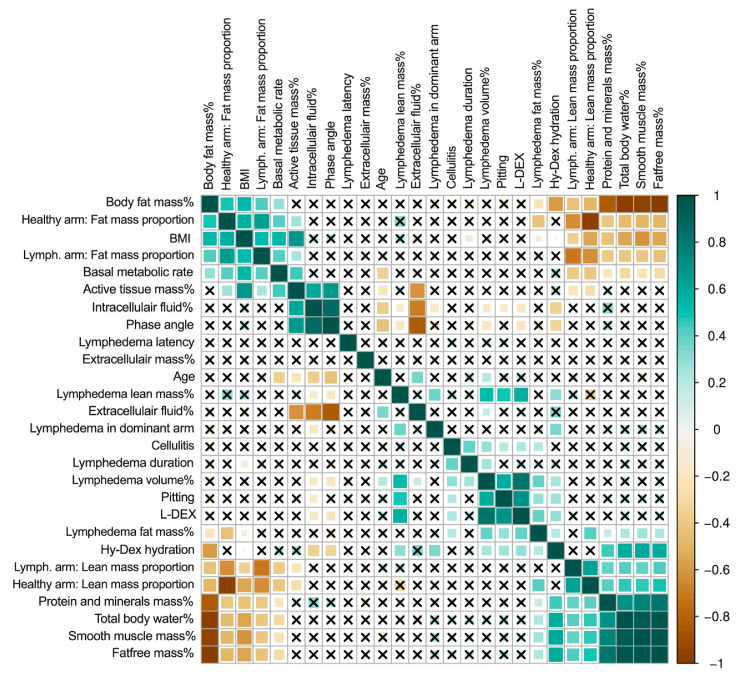
Correlation matrix of lymphedema and patient variables. This figure shows the correlation matrix between all variables. The darker green color denotes a stronger positive correlation, and the darker brown color denotes a stronger negative correlation. The squares with crosses denote that there is no significant correlation. Body fat mass %, basal metabolic rate, phase angle, intracellular fluid %, extracellular fluid %, extracellular mass %, total body water, protein and minerals mass %, Hy-Dex hydration, smooth muscle mass %, total body water %, active tissue mass%, and fat-free mass % are derived from whole-body analysis using bioimpedance. Lymphedema volume %, lymphedema lean mass %, and lymphedema fat mass %, lymph.arm: lean mass proportion, lymph.arm: fat mass proportion, healthy arm: lean mass proportion, healthy arm: fat mass proportion are regional analysis derived from DEXA scans. BMI = body mass index. Lymph. = lymphedema.

**Figure 3 cancers-13-03584-f003:**
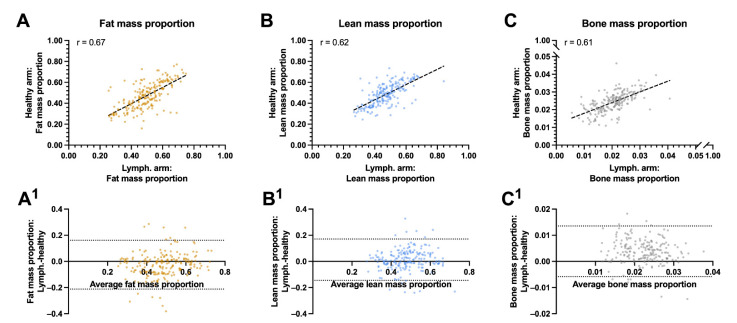
Correlation and agreement between biocomposition proportions between the lymphedema and healthy arms: This figure shows the correlation and agreement between the lymphedema and healthy arms proportion of fat, lean and bone mass. (**A**) Fat mass to whole arm ratio between the lymphedema and healthy arm. (**A^1^**) Agreements between the fat mass proportion of the lymphedema and healthy arm. (**B**) Lean mass to whole arm proportion between the lymphedema and healthy arm. (**B^1^**) Agreements between the lean mass proportion of the lymphedema and healthy arm. (**C**) bone mass to whole arm proportion between the lymphedema and healthy arm. (**C^1^**) Agreements between the bone mass proportion of the lymphedema and healthy arm. The whole arm proportion of lean, fat and bone mass correlated moderately between the lymphedema and healthy arm and there was a good agreement between the proportion of fat mass, lean mass and bone mass in the lymphedema and healthy arm. The striped lines shows the line of best fit in panel A, B and C. The horizontal dotted lines denote the 95% confidence intervals for the limits of agreements in panel A^1^, B^1^, and C ^1^. R = pearson correlation.

**Figure 4 cancers-13-03584-f004:**
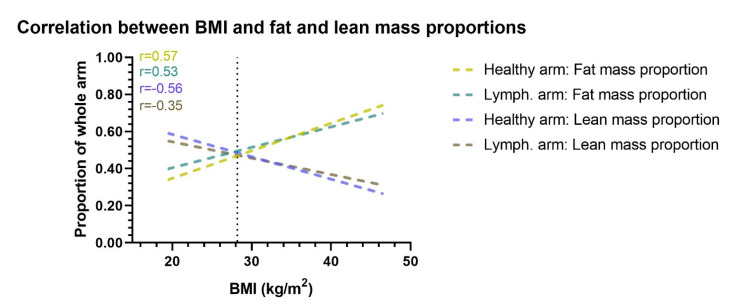
Correlation between BMI and arm proportion of fat and lean mass. This figure shows the correlation between BMI and the proportional amount of lean and fat mass in the healthy and lymphedema arm. Correlation lines are shown as lines of best fit through a scatterplot. When BMI increases the lymphedema and healthy arms proportion of fat mass increases, while the proportion of lean mass decreases. The vertical dotted line denotes the mean BMI of the cohort. Lymph. = lymphedema. r= Pearson r correlation.

**Figure 5 cancers-13-03584-f005:**
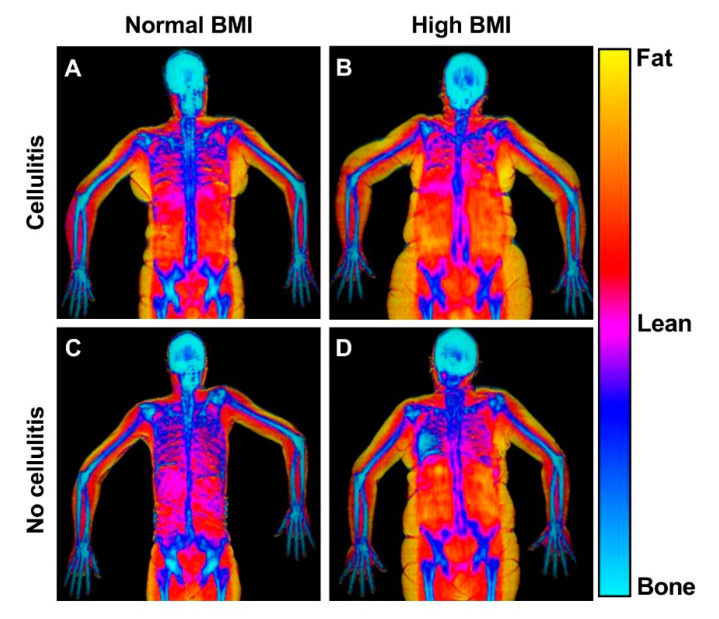
DEXA scans of patients with and without high BMI and previous cellulitis. This figure shows representative frontal plane DEXA scans of patients with normal and high BMI and with and without cellulitis. The DEXA scans reveal the bio composition of the lymphedema (right) and healthy (left) arms. (**A**) Patient with previous cellulitis, a BMI of 24 and 56% swelling. The healthy arm comprised of 29% fat, 64% lean mass and the lymphedema arm comprised of 27% fat, and 65% lean mass. (**B**) Patient with previous cellulitis, a BMI of 34 and with a 35% swelling. The healthy arm comprised of 54% fat, 43% lean mass and the lymphedema arm comprised of 59% fat, and 39% lean mass. (**C**) Patient with no previous cellulitis and a BMI of 22 and 20% swelling. The healthy arm comprised of 42% fat, 52% lean mass and the lymphedema arm comprised of 44% fat, and 51% lean mass. (**D**) Patient with no previous cellulitis and a BMI of 34. The healthy arm comprised of 44% fat, 51% lean mass and the lymphedema arm comprised of 45% fat, and 50% lean mass.

**Table 1 cancers-13-03584-t001:** Demographic and characteristics of included patients with and without cellulitis.

		All patients (*n* = 206)	No Cellulitis (*n* = 137)	Cellulitis (*n* = 69)	*p* Value
Age (years)	Mean ± SD	59.47 ± 10.05	58.87 ± 10.45	60.63 ± 9.17	n.s
BMI (kg/m^2^)	Median (IQR)	27.82 (7.13)	28.55 (6.96)	26.83 (7.48)	n.s
Breast cancer treatment variables					
Radiation therapy	N (%)	193 (93.69%)	128 (93.43%)	63 (94.20%)	n.s
Chemotherapy	N (%)	171 (83.41%)	112 (82.35%)	59 (85.51%)	n.s
Endocrine therapy	N (%)	164 (80.79%)	109 (81.34%)	55 (79.71%)	n.s
Mastectomy	N (%)	108 (52.43%)	69 (50.36%)	39 (56.52%)	n.s
Post-mastectomy reconstruction	N (%)	48 (44.44%)	29 (42.03%)	19 (48.72%)	n.s
Abdominal free flap	N (%)	21 (19.44%)	13 (18.84%)	8 (20.51%)	n.s
Pedicled back flap	N (%)	16 (14.81%)	13 (15.94%)	5 (12.82%)	n.s
Implant	N (%)	12 (11.11%)	6 (8.70%)	6 (15.38%)	n.s
Lymph nodes removed (No.)	Median (IQR)	17 (7)	17 (7)	17 (7)	n.s
Clinical lymphedema characteristics					
Lymphedema latency (years)	Median (IQR)	0.67 (1.43)	0.66 (1.35)	0.72 (1.79)	n.s
Lymphedema duration (years)	Median (IQR)	4.40 (5.52)	3.75 (4.42)	7.24 (6.57)	<0.001
Clinical pitting edema	N (%)	113 (54.85%)	65 (47.45%)	48 (69.57%)	<0.05
Dominant arm affected	N (%)	98 (47.57%)	61 (44.53%)	37 (53.62%)	n.s
Compression sleeve	N (%)	181 (87.86%)	118 (86.13%)	63 (91.30%)	n.s
Compression gauntlet	N (%)	125 (60.68%)	85 (62.04%)	40 (57.97%)	n.s
Night compression	N (%)	64 (31.07%)	36 (26.28%)	28 (40.58%)	<0.05
Pneumatic compression device	N (%)	40 (19.42%)	20 (14.60%)	20 (28.99%)	<0.05
Dual Energy X-Ray Absorptiometry analysis					
Lymphedema excess volume (%)	Median (IQR)	21.58 (29.03)	14.36 (26.31)	30.60 (27.82)	<0.001
Lymphedema excess lean mass (%)	Median (IQR)	13.57 (36.41)	9.13 (30.48)	21.30 (37.57)	<0.05
Lymphedema excess fat mass (%)	Median (IQR)	23.33 (39.55)	20.51 (35.57)	37.48 (50.53)	<0.05
Lymphedema excess bone mass (%)	Median (IQR)	−2.00 (30.21)	−3.37 (28.86)	2.03 (35.10)	n.s
Lymphedema arm: Fat mass proportion	Median (IQR)	0.50 (0.15)	0.50 (0.15)	0.48 (0.12)	n.s
Healthy arm: Fat mass proportion	Median (IQR)	0.47 (0.19)	0.49 (0.19)	0.45 (0.18)	n.s
Lymphedema arm: Lean mass proportion	Median (IQR)	0.47 (0.12)	0.46 (0.13)	0.48 (0.11)	n.s
Healthy arm: Fat mass/arm ratio	Median (IQR)	0.48 (0.16)	0.47 (0.16)	0.50 (0.15)	n.s
Bioimpedance spectroscopy analysis					
L-DEX	Median (IQR)	21.90 (29.50)	18.9 (28.50)	30.2 (29.40)	<0.001
Fat mass %	Median (IQR)	32.52 (9.00)	33.24 (9.07)	31.04 (7.71)	n.s
Protein and minerals mass %	Median (IQR)	18.07 (2.49)	17.89 (2.42)	18.40 (2.31)	n.s
Total body water %	Median (IQR)	49.75 (6.80)	49.34 (6.83)	50.59 (5.87)	n.s
Intracellular fluid %	Median (IQR)	54.64 (2.47)	54.64 (2.53)	54.67 (2.39)	n.s
Extracellular fluid %	Median (IQR)	45.31 (2.43)	45.32 (2.57)	45.25 (2.34)	n.s
Fat-free mass %	Median (IQR)	67.50 (9.19)	66.97 (9.08)	68.96 (8.05)	n.s
Active tissue mass %	Median (IQR)	35.24 (5.18)	35.03 (5.22)	35.91 (5.00)	n.s
Extracellular mass %	Median (IQR)	32.35 (5.07)	32.08 (5.04)	33.36 (4.71)	n.s
Smooth muscle mass %	Median (IQR)	27.09 (5.22)	26.76 (5.52)	27.88 (4.46)	n.s
Basal metabolic rate (cals/day)	Median (IQR)	1336.30 (235.25)	1355.51 (226.50)	1319.82 (256.00)	n.s
Phase angle (°)	Median (IQR)	4.50 (0.90)	4.50 (0.70)	4.45 (0.90)	n.s
Hy-DEX hydration analysis	Median (IQR)	13.95 (23.5)	12.55 (19.4)	16.5 (27.4)	n.s
Indocyanine green lymphangiography					
MD Anderson stage (median stage)	Median (IQR)	3 (2)	2 (2)	3 (1)	<0.05

Table 1 legend: This table shows the demographics of the included patients. IQR = interquartile range. n.s = not significant. Comparisons between patients with and without cellulitis were compared using unpaired t-test, Chi-squared, or Mann–Whitney test depending on data type and distribution.

**Table 2 cancers-13-03584-t002:** Multivariate analysis for lymphedema volume, fat mass and lean mass.

Variables	Lymphedema Volume (%)		Lymphedema Fat Mass (%)		Lymphedema Lean Mass (%)	
	Coefficient (95% CI)	*p*-Value	Coefficient (95% CI)	*p*-Value	Coefficient (95% CI)	*p*-Value
Age (per age)	−0.01 (−0.29; 0.26)	n.s	−0.45 (−1.14; 0.25)	n.s	0.19 (−0.02; 0.59)	n.s
BMI (per point)	1.44 (0.74; 2.13)	<0.001	0.69 (-1.12; 2.50)	n.s	2.18 (1.21; 3.15)	<0.001
Lymphedema duration (per year)	0.18 (−0.51; 0.88)	n.s	0.26 (−1.48; 2.00)	n.s	0.31 (−0.67; 1.29)	n.s
Cellulitis (yes)	12.35 (6.23; 18.47)	<0.001	20.74 (5.35; 36.12)	<0.001	9.95 (1.28; 18.62)	<0.05
Bodily smooth muscle mass (per 1%)	1.77 (−0.55; 3.00)	n.s	2.99 (−0.07; 6.05)	n.s	1.66 (−0.05; 3.38)	n.s
Bodily fat mass (per 1%)	−0.11 (−0.87; 0.65)	n.s	0.16 (−1.72; 2.05)	n.s	0.07 (−0.98; 1.12)	n.s
Lymphedema in dominant arm (yes)	13.18 (7.62; 18.73)	<0.001	8.76 (−5.67; 23.18)	n.s	17.36 (9.62; 25.10)	<0.001

Table 2 legend: This table shows the multivariate linear regression models for lymphedema volume, fat mass, and lean mass predictors. n.s = not significant.

**Table 3 cancers-13-03584-t003:** Multivariate analysis of the lymphedema and healthy arms proportions of lean and fat mass.

Variables	Lymph. Arm: Lean Mass Proportion		Healthy Arm: Lean Mass Proportion		Lymph. Arm: Fat Mass Proportion		Healthy Arm: Fat Mass Proportion	
	Coefficient (95% CI)	*p*-Value	Coefficient (95% CI)	*p*-Value	Coefficient (95% CI)	*p*-Value	Coefficient (95%CI)	*p*-Value
Age (per age)	0.01 (0.00; 0.01)	<0.05	0.00(−0.00; 0.01)	n.s	−0.01(−0.01; −0.00)	<0.05	−0.00 (−0.01; 0.00)	n.s
BMI (per point)	−0.02 (−0.04; −0.01)	<0.001	−0.04 (−0.05; −0.03)	<0.001	0.03 (0.02; 0.05)	<0.001	0.05 (0.03; 0.06)	<0.001
Lymphedema duration (per year)	0.00 (−0.02; 0.01)	n.s	−0.00 (−0.02; 0.01)	n.s	0.00 (−0.01; 0.02)	n.s	0.01 (−0.01; 0.02)	n.s
Cellulitis (yes)	−0.02 (−0.12; 0.09)	n.s	0.03 (−0.07; 0.14)	n.s	0.01 (−0.11; 0.13.)	n.s	−0.06 (−0.19; 0.07)	n.s
Bodily smooth muscle mass (per 1%)	0.00 (−0.02; 2.10)	n.s	−0.00 (−0.02; 0.02)	n.s	0.00 (−0.02; 0.03)	n.s	−0.00 (−0.03; 0.02)	n.s
Body fat mass (per 1%)	−0.02 (−0.03; -0.00)	<0.05	−0.02 (−0.03; −0.00)	<0.05	0.02 (0.00; 0.03)	<0.05	0.02 (0.00; 0.03)	<0.05
Lymphedema in dominant arm (yes)	0.09 (−0.08;0.26)	n.s	0.03 (−0.14; 0.20)	n.s	−0.11 (−0.29; 0.08)	n.s	−0.04 (−0.24; 0.17)	n.s

Table 3 legend: This table shows the multivariate beta regression models for fraction of arm mass comprised of fat and lean mass in the healthy and lymphedema arm. Increase in BMI and body fat mass was associated with a higher fraction of fat and lower fraction of lean mass in both the healthy and lymphedema arm. N.s = not significant. Lymph = lymphedema. N.s = not significant.

## Data Availability

The data presented in this study are available on request from the corresponding author upon reasonable request.

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
