# Peer review of "Cellulitis Is Associated with Severe Breast Cancer-Related Lymphedema: An Observational Study of Tissue Composition"

_cancers, 2021, doi:10.3390/cancers13143584_

Round 1
Reviewer 1 Report
After carefully reading the concerns of the reviewers a d the response of the authors I can state that the authors carefully followed all the suggestions of the review process.
A minor question has been raised to me whether the authors did not mention radiation therapy of the axillary region, or did I miss it?
Generally I suggest publication of this interesting and important study.
Reviewer 2 Report
We congratulate the authors for selecting and presenting the data.
The authors should comment or change the following:
In Keywords:
Line 35: the word “microsurgery” must be deleted, since the study has no microsurgical implication
In Introduction:
Line 41: the word “abruption” does not reflect to the meaning of the disease. I would suggest, to be changed with “obstruction” or any similar.
In Materials and Methods:
The “Indocyaninegreen” in the whole text must be divided in two words as “Indocyanine green”; it is written correct in line 100.
In Results:
The findings of the Indocyanine Green Lymphangiography are not well presented, neither the number of patients that underwent the examination. In Table 1, it is rather confusing the numbers of patient who had and the ICG, or the mean stage? Could you please clarify?
In Discussion:
Line 390: “power-assisted” to be deleted (just liposuction is enough), since the reference article do not describe power-assisted liposuction either.
In Conclusion:
Line 488: “Specifically, weight optimization should be encouraged for BCRL patients and surgical treatments such as lymphovenous bypass and liposuction should be individualized or combined based on lymphedema bio composition.”
Since the present study does not analyze surgical procedures and factors, the above conclusion can not be made for this study (even if my personal opinion consorts with).
Author Response
Please see the attachment

This manuscript is a resubmission of an earlier submission. The following is a list of the peer review reports and author responses from that submission.
Round 1
Reviewer 1 Report
The authors investigated the association between a previous episode of cellulite and the amount of fat mass in lymphedema in patients with Breast Cancer-Related Lymphedema through Dual Energy X-Ray Absorptiometry (DEXA) scans and bioimpedance spectroscopy analyzes. The authors also evaluated what factors may be associated with a higher proportion of fat mass in the lymphedema arm.
The manuscript is clear and well organized. The methods and results are clearly set out. Here are just a few considerations to improve the clarity of the manuscript.
1) It would probably be useful to supplement the studies at the state of the introduction in the introduction.
2) In the paragraph 'Excess arm mass' the p-values ​​relating to the associations found to be significant or at least the level of significance considered, presumably 0.05, are not reported.
3) same previous consideration for the correlation results reported in the paragraph 'Fat and lean arm mass proportions'
4) in the discussions the authors state that 'This is the first study investigating the impact of cellulitis and body composition on lymphedema bio composition.' Nonetheless, there is no framing of the current state-of-the-art studies worthy of note that should be discussed,
- Asdourian, M. S., Skolny, M. N., Brunelle, C., Seward, C. E., Salama, L., & Taghian, A. G. (2016). Precautions for breast cancer-related lymphoedema: risk from air travel, ipsilateral arm blood pressure measurements, skin puncture, extreme temperatures, and cellulitis. The lancet oncology, 17 (9), e392-e405.
- He, L., Qu, H., Wu, Q., & Song, Y. (2020). Lymphedema in survivors of breast cancer. Oncology letters, 19 (3), 2085-2096.
5) Predicting the involvement of the lymph nodes could help to avoid surgery on the armpit, even of a minimally invasive type, such as sentinel lymph node biopsy, especially in patients with extensive expression of fat mass in the upper limb and previous episodes of cellulite and to mention the work. In fact, a review published on CA: A Cancer Journal for Clinicians, a group of researchers from the University of Texas M.D. Anderson Cancer Center, offers an in-depth overview of knowledge on the subject, reviewing the scientific literature on the subject and analyzing the incidence rates of lymphedema in the most common forms of cancer, with particular attention to techniques for visualizing and measuring the pathology. Their work compares the results of several clinical studies that have dealt with describing the presence of lymphedema in patients with breast cancer, reporting an average incidence in patients following sentinel lymph nodes biopsy (SLNB) equal to 6.3% and 22.3% after axillary lymph node removal (ALND). The definition of less invasive techniques of sentinel lymph node biopsy could allow a reduction of the problem in clinically negative patients. (Fanizzi, A., Pomarico, D., Paradiso, A., Bove, S., Diotaiuti, S., Didonna, V., ... & Massafra, R. (2021). Predicting of Sentinel Lymph Node Status in Breast Cancer Patients with Clinically Negative Nodes: A Validation Study. Cancers, 13(2), 352. )
Reviewer 2 Report
- Introduction
- The aims of the study were stated but not consistent throughout the rest of the article (Please see comment 5-1).
- More information is needed the to summarize the current state of the topic (cellulitis in BCRL) and to establish the need for this study. Perhaps the introduction is not enough..
- Materials and Methods
- The inclusion criteria were not properly defined.
- How was lymphedema diagnosed? Were there circumference or volume measurements? Was lymphoscintigraphy performed?
- What is ASA 1 or 2? There was no definition of the abbreviation.
- The definitions of terms are not clear.
- This study focuses on cellulitis but “cellulitis” was not properly defined.
- Certain parameters in the outcome are not familiar or may not be appropriate, such as
- Active tissue mass: Please provide definition.
- Hy-Dex hydration score: Please provide definition. How is this obtained?
- For the Variables part (Lines 174-199)
- The aims of the study were mentioned again. This is redundant and not appropriate for the Methods section. Please rephrase.
- Are the parameters derived from the SOZOpro software reliable? Are there enough references to support this?
- The inclusion criteria were not properly defined.
- Results and Discussion
- For the bioimpedance spectroscopy parameters, only L-DEX score showed a significant difference between those with previous episode of cellulitis and those without.L-DEX was almost twice as high in the former compared to the latter, however, no other difference was noted in other parameters. How can you explain this? Since high L-DEX (>10) as mentioned in the Methods signifies more fluid, how can you explain the lack of difference in the amount of extracellular fluid or Hy-Dex hydration between the 2 groups? Can you use the information of ECF in the each limb?
- In Table 1, why are some parameters in italics? Is this necessary? Also in Figure 2, some parameters are in bold. What does this signify?
- In Table 2 legend, indocyanine green stage was mentioned.However, this was not mentioned in the Methods and in other parts of the paper.
- Conclusion
- The conclusion does not clearly answer the objectives of the study.
- Lines 413-415: Evidence from this study implies that lymphedema bio composition is diverse for each patient and is associated with previous cellulitis episodes, BMI, body fat mass and arm dominance.
- The primary aim was to investigate the association between cellulitis and fat mass. However, the statement above focuses on the association of lymphedema bio-composition with cellulitis and other parameters.Please rephrase this statement. Which should come first? Cellulitis or lymphedema bio-composition?
- Lines 413-415: Evidence from this study implies that lymphedema bio composition is diverse for each patient and is associated with previous cellulitis episodes, BMI, body fat mass and arm dominance.
- The conclusion does not clearly answer the objectives of the study.
- The authors need to improve on the coherence or consistency on the use of terms throughout the different parts of the paper.
- Since this is an observational study, this study can only show the association and not causality between cellulitis and BCRL.However, several inconsistencies were noted:
- Title: “…impact of cellulitis...” (Line 3): the term “impact” denotes causality
- Simple summary: “…where cellulitis affect the severity…” (line 17)” the term “affect” also reflects causality; vs “Cellulitis was associated with…” (line 18)
- Since this is an observational study, this study can only show the association and not causality between cellulitis and BCRL.However, several inconsistencies were noted:
- Introduction: “The primary aim was to investigate the association…”
- Conclusion: “…lymphedema bio-composition…is associated with…”
- Figure 3 and 4: The terms “proportion” and “ratio” were used interchangeably.Though related they are different entities, please clarify which one was used and be consistent throughout.
Reviewer 3 Report
Thank you for this thoughtful and very well presented study which confirms what seems to be the case clinically - that patients with BCRL and a history of cellulitis seem to have "fatter" arms. It is interesting that duration of BCRL was not important (at least in this group) and seems to tell us that patients who experience cellulitis differ in some important way from those who don't - with cellulitis perhaps being a symptom of some other variable (e.g. lymphatic pulsatile flow?) just as fat deposition is a symptom of cellulitis. Exercise has been shown to reduce the likelihood of lymphedema and one would think this would be an advantage for the dominant arm since it tends to be used more for lifting. One wonders what activities of the dominant arm are the problem (e.g. is it more likely to sustain minor injuries that predispose to infection). You are not able to determine whether cellulitis is associated with fat deposition or fat deposition predisposes to cellulitis. Are fat women more likely to get cellulitis? That would seem to be worth answering. I have no specific suggestions or criticisms. Congratulations.
Reviewer 4 Report
Thank you for allowing me to review this paper. The authors investigated the association between a previous cellulites episode and the amount of fat mass. The paper is really interesting and worthy for publications. Actually, I do not like the title of the paper suggesting a more informative title. For instance, something like “The association between lymphedema, previous cellulites episode and the amount of fat mass ”. I have not other major comment.